# E$^2$CM: Early Exit via Class Means for Efficient Supervised and Unsupervised Learning

## Abstract

State-of-the-art neural networks with early exit mechanisms often need considerable amount of training and fine-tuning to achieve good performance with low computational cost. We propose a novel early exit technique, E$^2$CM, based on the class means of samples. Unlike most existing schemes, E$^2$CM does not require gradient-based training of internal classifiers. This makes it particularly useful for neural network training in low-power devices, as in wireless edge networks. In particular, given a fixed training time budget, E$^2$CM achieves higher accuracy as compared to existing early exit mechanisms. Moreover, if there are no limitations on the training time budget, E$^2$CM can be combined with an existing early exit scheme to boost the latter's performance, achieving a better trade-off between computational cost and network accuracy. We also show that E$^2$CM can be used to decrease the computational cost in unsupervised learning tasks.

## 1 Introduction

Modern deep learning models require a vast amount of computational resources to effectively perform various tasks such as object detection (Szegedy et al., 2015), image classification (He et al., 2016), machine translation, (Vaswani et al., 2017) and text generation (Brown et al., 2020). Deploying deep learning models to the edge, such as to mobile phones or the Internet of Things (IoT), thus becomes particularly challenging due to device computation and energy limitations (Li et al., 2021a;b). Moreover, the law of diminishing returns applies to the computation-performance trade-off (Bolukbasi et al., 2017): The increase in a deep learning model's performance is often marginal as compared to the increase in the amount of computation.

One of the primary reasons behind traditional deep learning models' high computation demand is their tunnel-like design. In fact, traditional models apply the same sequence of operations to any given input. However, in many real world datasets, certain inputs may consist of much simpler features as compared to other inputs (Bolukbasi et al., 2017). In such a scenario, it becomes desirable to design more efficient architectures that can exploit the heterogeneous complexity of dataset members. This can be achieved by introducing additional exit points to the models (Panda et al., 2016; Teerapittayanon et al., 2016; Kaya et al., 2019). These exit points prevent simple inputs to traverse the entire network, reducing the computational cost of inference.

Despite reduced inference time, existing early exit neural network architectures require additional training and fine-tuning for the early exit points, which increases the training time (Teerapittayanon et al., 2016; Bolukbasi et al., 2017; Kaya et al., 2019). This side-effect is undesirable for scenarios in which the training has to be done in a low-power device. An ideal solution is a plug-and-play approach that does not require gradient-based training and performs well. In this work, we propose such an early-exit mechanism, Early Exit Class Means (E$^2$CM), based on the *class means* of input samples for the image classification task. By taking the mean of layer outputs for each class at every layer of the model, class means are obtained. During inference, output of a layer is compared with the corresponding class means using Euclidean distance as the metric. If the output of the layer is close enough to a class mean, the execution is stopped and the sample exits the network. In fact, as seen in Fig. 1, some samples can be classified easily at early stages of the network by just considering a "nearest class mean" decision rule, suggesting the potential effectiveness of our method for reducing computational cost.

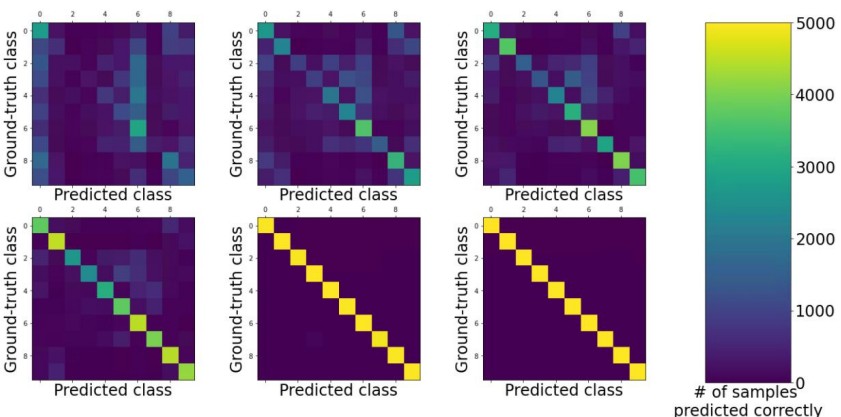

Figure 1: Confusion matrix of the classifications done according to the nearest class mean on CIFAR-10 training set. From left to right, top to bottom; the results belong to the first convolutional layer, $1^{st}$, $3^{rd}$, $10^{th}$, $15^{th}$ and $30^{th}$ residual block of ResNet-152.

A practical use case for $E^2CM$ is when a large, expensive-to-train model is broadcast to edge devices with limited and heterogeneous computation capabilities. In such a scenario, different devices may train the $E^2CM$ model at different FLOP operating points depending on their computation capabilities or power limitations. Another practical scenario for $E^2CM$ may be transfer learning, where the training has to be done on a low power edge device with a local dataset that is different from the dataset the base network was trained on. In order to keep the battery usage minimum on the edge device, the training time may be limited. The advantage of $E^2CM$ is that it does not need any gradient-based training like Shallow-Deep Networks (Kaya et al., 2019), therefore it is more suitable for transfer learning on low power devices.

One other practical scenario for $E^2CM$ may be transfer learning, where the training has to be done on low power edge device with a local dataset that is different from the dataset the base network was trained on. In order to keep the battery usage minimum on the edge device, the training time may be limited. The advantage of $E^2CM$ is that it does not need any gradient-based training like Shallow-Deep Networks (Kaya et al., 2019).

To the best of our knowledge, $E^2CM$ is the first early exit mechanism that does not require a gradient-based training and does not modify the original network by any means. Furthermore, it does not have a hyper-parameter for the early exit locations unlike existing schemes. These features make $E^2CM$ simpler to use and easier to deploy on low-power devices. While using class means as the only early exit mechanism requires just a single feed forward pass, existing early exit methods reaches the same performance after training for multiple epochs, which suggests our method is more agile and powerful yet simpler. Moreover, combining $E^2CM$ with the existing mechanisms that require gradient-based training achieves a better trade-off in terms of computation cost and network accuracy. In addition to its benefits in the supervised learning setting, $E^2CM$ is also the first technique that shows the feasibility of early exits in the unsupervised learning setting.

We show the effectiveness of $E^2CM$ on CIFAR-10, CIFAR-100 (Krizhevsky et al., 2009) and KMNIST (Clanuwat et al., 2018) datasets using ResNet-18, ResNet-152 (He et al., 2016) and WideResNet-101 models (Zagoruyko & Komodakis, 2016). Using class means as the sole decision mechanism for early exiting results in 35% better accuracy or 50% faster inference time compared to the current state-of-the-art early exit technique. When combined with the state-of-the-art, we increase the accuracy by 9% without doing further computation; or decrease the inference time by 25% without a loss in accuracy. We also show that it is possible to decrease the computational cost while doing clustering with autoencoders on MNIST (LeCun et al., 1998). In particular, $E^2CM$ saves on computation by 60%, while the loss in unsupervised clustering accuracy (Xie et al., 2016) is marginal.

## 2 RELATED WORK

Our work is related to the area of *conditional computation* (Bengio et al., 2013), where several small networks are trained to control the computation flow of one deep neural network. For this purpose, adding gates between the blocks of residual networks have been proposed (Veit & Belongie, 2018; Wang et al., 2018). During inference, these gates allow the input to skip unnecessary blocks, thus saving computation time. However, unlike our proposed method, the gated network has to be trained from scratch. Conditional computation policies can also be learned through reinforcement learning (Bengio et al., 2015; Jie et al., 2019). However, this latter approach forces the inputs to go through the entire network as it does not incorporate early exit points (Bengio et al., 2015).

One of the earliest works that explicitly propose the idea of early exiting is Panda et al. (2016), where the authors consider adding a cascade of linear layers after convolutional layers as control blocks. Rather than just linear layers, adding *branches* consisting of convolutional layers to the original model has also been studied (Teerapittayanon et al., 2016). A significant drawback of this method is that the branches may increase the computational cost due to convolutional layers. In addition, this idea requires branches to be trained with the original model jointly, from scratch. In a more recent study, *internal classifiers (ICs)* consisting of a feature reduction layer and a single linear layer are added after certain layers in the network (Kaya et al., 2019). Hence, the methods presented in these studies modify the original network by adding linear or convolutional layers. Moreover, they require gradient updates to train those layers. Also, an implicit hyper-parameter is the locations of the early exit points. E$^2$CM is better suited for low-power applications compared to existing studies since we do not modify the original model, do not require gradient based training and additional hyper-parameters.

In addition to layer level early exits, a network level early exit mechanism has been introduced in Bolukbasi et al. (2017). Both the layer level exit and the network level exit require decision functions to be inserted between the layers and networks. This type of architecture freezes the weights of the original model, and then trains the decision functions one by one using weighted binary classification. The drawback of this approach is such an alternative optimization, which may consume a lot of time and energy when there are many decision functions to optimize.

The idea of early exit neural networks shows promising results in natural language processing domain too. Adding early exit points between the transformer layers reduce the computation time with marginal loss in performance (Zhou et al., 2020; Xin et al., 2020). Applications for neural network based early exit mechanisms to computer fault management have also been considered (Biasielli et al., 2020).

Intermediate layer outputs have been used for classification in few-shot and one-shot learning settings in the past (Koch et al., 2015; Vinyals et al., 2016; Snell et al., 2017; Sung et al., 2018). These studies are closely related with the area of *metric learning*. The closest work to E$^2$CM is *prototypical networks*, in which *prototypes* for each class are computed (Pan et al., 2019). However, none of those approaches aim to reduce the computation cost.

E$^2$CM is also related to the phenomenon of neural collapse (Papyan et al., 2020). It is known that as the inputs go deeper in a neural network, the classes are separated better from each other as a result of multiple nonlinearities, and the samples begin to concentrate (Cohen et al., 2020; Zarka et al., 2020). E$^2$CM exploits this phenomenon with the main idea of stopping the execution as soon as the sample is close enough to a concentration point, i.e., a class mean.

One other idea to reduce the computational cost of neural networks during inference time in the literature is the usage of multiresolution inputs and networks (Huang et al., 2017; Yang et al., 2020; McGill & Perona, 2017). While this idea works well, it needs subnetworks and the usage of multiresolution inputs, which means it requires more hyperparameters compared to E$^2$CM.

## 3 EARLY EXIT CLASS MEANS

We study the problem of image classification. Let $(x_0^{(i)}, y^{(i)}) \in D$ be an image-label pair from the dataset $D$ consisting of $N$ samples and $K$ distinct classes, where $y^{(i)} \in \{1, 2, \ldots, K\}$ and $i \in \{1, 2, \ldots, N\}$. We denote the network $F$ with $M$ layers as a sequence $l_1, l_2, \ldots, l_M$. Let $\hat{y}^{(i)}$

denote the prediction of the network, $x_j^{(i)}$ denote the output of layer $j$, and $\hat{y}_j^{(i)}$ denote the prediction in case the input exits the network after layer $j$, for $j = 1, 2, \ldots, M$. We can write the full equation for the network $F$ as

$$x_j^{(i)} = l_j(x_{j-1}^{(i)}), \, j = 1, 2, \ldots, M. \tag{1}$$

## 3.1 CLASS MEANS

The input to E$^2$CM is the network $F$ trained on $D$. The network $F$ is not modified by any means. Therefore, we can obtain the class means for each class at each layer easily by just a forward pass. This is especially useful when the training time budget is fixed. Let $S_k$ denote the set of samples whose ground-truth label is $k$, and $c_j^k$ denote the mean of the output of layer $j$ for class $k$. In other words, let

$$c_j^k = \frac{1}{|S_k|} \sum_{n \in S_k} x_j^{(n)}. \tag{2}$$

Then, the Euclidean distance between a layer output $x_j^{(i)}$ and $K$ class means $c_j^k$ is computed via

$$d_j^{k(i)} = ||x_j^{(i)} - c_j^k||_2, \, k \in \{1, 2, \ldots, K\}. \tag{3}$$

After calculating $d_j^{k(i)}$ at each layer for every sample in the dataset, we normalize the distances for each class as

$$d_j^{k(i)} := \frac{d_j^{k(i)}}{\frac{1}{N} \sum_{i=1}^{N} d_j^{k(i)}}, \, k \in \{1, 2, \ldots, K\}. \tag{4}$$

Finally, the normalized distances are converted to probabilities of input belonging to a class in order to perform inference. This is done using the softmax function as

$$P(\hat{y}_j^{(i)} = k) = \text{softmax}(-d_j^{k(i)}). \tag{5}$$

During inference, the decision of exiting after $l_j$ or moving forward to $l_{j+1}$ is made according to a threshold value $T_j$. If the largest softmax probability is greater than the specified threshold $T_j$, execution is stopped and the class with the largest softmax probability is predicted. In other words, if

$$\max(\text{softmax}(-d_j^{k(i)})) > T_j, \tag{6}$$

then the network predicts

$$\hat{y}_j^{(i)} = \arg\max_k(\text{softmax}(-d_j^{k(i)})). \tag{7}$$

Otherwise, the input moves forward to the next layer. In the worst case, execution ends at the last layer of the network.

E$^2$CM performs differently according to different set of threshold values. We randomly initialize the thresholds and use binary search updates to reach the target number of FLOPs on training set. Later, same threshold values are used on the test set during the inference phase for that target number of FLOPs. The full procedure of our method is shown in Algorithm 1.

## 3.2 COMBINATION OF CLASS MEANS WITH EXISTING SCHEMES

Performance of an existing early exit scheme can be boosted if they are combined with E$^2$CM. In this context, existing methods decide either according to the entropy of the early exit prediction (Teerapittayanon et al., 2016), or the largest probability value in the prediction (Kaya et al., 2019). In the former strategy, the input to the neural network exits early if the entropy is smaller than a threshold $T_j$ for Layer $j$ of the network. In the latter, the sample exits early if the largest probability value is greater than $T_j$.

Existing methods use only $x_j^{(i)}$ as the input to the next layer. We propose feeding the class means $c_4^k j$ as additional inputs to the layers by simple concatenation, which improves the performance. During inference, if the $j^{th}$ internal classifier decides to move to the next layer, class means are consulted. If class means do not approve moving on, the prediction of the $j^{th}$ internal classifier is returned and the input exits early. Hence, an input can move to the next layer if and only if it receives approval from both the internal classifier (which can be based on any existing scheme) and our simple E$^2$CM.

---

**Algorithm 1** Early Exit Class Means (E$^2$CM)

---

**Input:** Trained network layers $l_j$, dataset $D$, thresholds $T_j$
**if** training **then**
    **for** $j = 1$ **to** $M$ **do**
        $x_j^{(i)} = l_j(x_{j-1}^{(i)})$
        Calculate class means $c_j^k$, $k \in \{1, 2, \ldots, K\}$
    **end for**
**end if**
**if** inference **then**
    **for** $j = 1$ **to** $M$ **do**
        $x_j^{(i)} = l_j(x_{j-1}^{(i)})$
        Compute $d_j^{k^{(i)}} = ||x_j^{(i)} - c_j^k||_2$
        Normalize $d_j^{k^{(i)}}$ as in (4).
        **if** $\max(\text{softmax}(-d_j^{k^{(i)}})) > T_j$ **then**
            Early exit with $\arg\max_k(\text{softmax}(-d_j^{k^{(i)}}))$.
        **end if**
    **end for**
**end if**

---

### 3.3 EXTENSION TO UNSUPERVISED LEARNING

E$^2$CM can be used for clustering as well. As an example, we consider Deep Embedding Clustering (DEC) (Xie et al., 2016) and focus on the task of jointly learning representations and cluster assignments. In DEC, there is only one clustering layer, and it is at the end of the encoder layers. This makes it impractical for low power clustering, because the architecture is like a tunnel with only one exit at the end. We propose adding multiple clustering layers as early exits in order to decrease the computational cost.

## 4 RESULTS

We validate the effectiveness of our method on CIFAR-10, CIFAR-100 (Krizhevsky et al., 2009), KMNIST (Clanuwat et al., 2018), and MNIST (LeCun et al., 1998) datasets. CIFAR-10 and CIFAR-100 dataset consists of 50000 training and 10000 test images, and have 10 and 100 classes respectively with equal amount of samples for each class. KMNIST dataset consists of 60000 training and 10000 test images like MNIST. While MNIST is a dataset of handwritten digits, KMNIST consists of 10 Hiragana characters, which are grayscale images unlike CIFAR-10 images. We use ResNet-18, ResNet-152 (He et al., 2016) and WideResNet-101 (Zagoruyko & Komodakis, 2016) models for our supervised learning experiments. For all models, we use the same data augmentation scheme and the hyper-parameter values stated in Veit & Belongie (2018) for training. We use cross-entropy loss for all trainings. For our unsupervised learning experiments, we use the same network architecture and training parameters stated in Xie et al. (2016).

We run experiments in three settings. First, we fix the training time budget and compare E$^2$CM with the existing early exit methods in terms of network accuracy and floating point operations (FLOPs) performed during inference. Euclidean distance calculations are included in FLOPs for our method. In the second setting, we lift the training time budget. We allow the full training of the internal classifiers, and we compare the combination of E$^2$CM and internal classifiers against existing methods. The third setting is for unsupervised learning, and we generate the accuracy-FLOPs curve to evaluate the effectiveness of E$^2$CM.

### 4.1 CLASS MEANS UNDER A FIXED TRAINING TIME BUDGET

In the fixed training time budget setting, we compare E$^2$CM with existing methods in two ways. First, E$^2$CM is compared with Shallow-Deep Networks (Kaya et al., 2019) and BranchyNet (Teerapittayanon et al., 2016), which are trained for only one epoch since E$^2$CM requires only a sin-

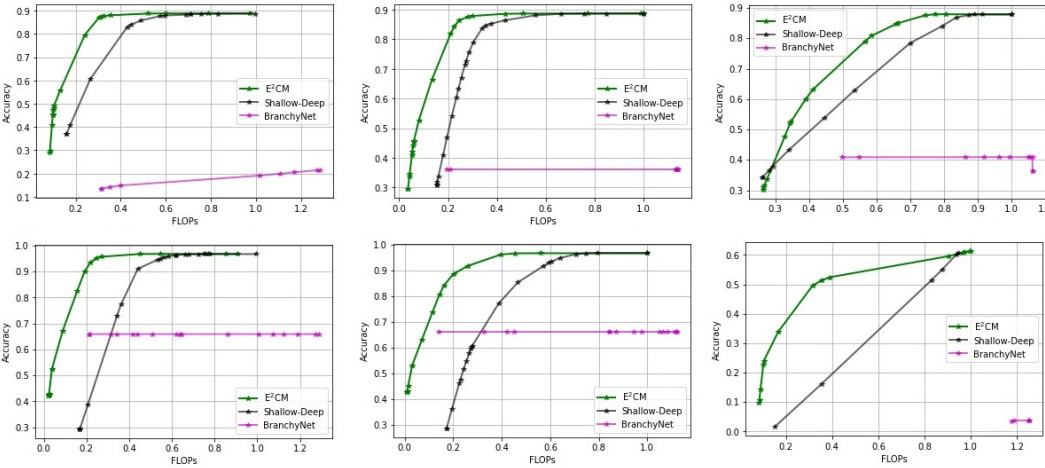

Figure 2: Comparison of E²CM with existing methods under fixed training time budget of one epoch for ResNet-152 (left) and WideResNet-101 (middle) on CIFAR-10 (top) and KMNIST (bottom). ResNet-18 for CIFAR-10 is shown at top right, and ResNet-152 for CIFAR-100 is shown at bottom right.

gle forward pass. We do not include here the Bolukbasi-Wang-Dekel-Saligrama (BWDS) method (Bolukbasi et al., 2017), because in this method, each decision function requires a separate training. Hence, we cannot train an entire BWDS network with many decision functions using only one epoch.

Shallow-Deep Networks add internal classifiers after certain layers in the network. We use the procedure described in Kaya et al. (2019) and add 6 ICs after the layers which correspond to the 15%, 30%, 45%, 60%, 75%, 90% of the entire network in terms of FLOPs. For BranchyNet, we add 2 branches to the original network. The first branch is after the first convolutional layer, and the second branch is after the layer that corresponds to 1/3 of the whole network in terms of FLOPs.

Secondly, we still consider a fixed training time budget, but this time we allow the separate training of decision functions in the BWDS method. As suggested by the authors, there are 6 early exit points, hence 6 decision functions. We train each decision function for 1 epoch, resulting in 6 separate epochs. To make a fair comparison, we train Shallow-Deep Networks and BranchyNet for 6 epochs as well. We combine E²CM with Shallow-Deep Networks. By doing so, E²CM rectifies the decisions made by Shallow-Deep Networks. At every exit point of Shallow-Deep Networks, if the decision is made in favor of exiting early at the exit point, E²CM is consulted. If E²CM disagrees with Shallow-Deep Networks, early exit does not happen at that exit point.

To reach a target number of FLOPs, we first initialize the threshold vector by drawing $M$ numbers uniformly at random from $[0, 1]$. The $j^{th}$ component of the threshold vector corresponds to $T_j$ in Algorithm 1 and is utilized at Layer $l_j$ of the neural network. Then, the softmax values are obtained and saved for each layer on training set as in Eq. 5. Therefore, only one pass on training set is needed to optimize the thresholds. We update the threshold vector until we reach the target number of FLOPs using binary search at each component and alternating optimization: if the thresholds give larger number of FLOPs than the target FLOP, the thresholds are decreased for the next iteration. Otherwise, if they give smaller number of FLOPs than the target FLOP, they are increased. Binary search is guaranteed to converge as the (average) FLOPS are monotonic with respect to the thresholds.

To obtain the overall tradeoff between accuracy and FLOPS, the above threshold optimization process is repeated on the training set for target FLOPs ranging from 0.0 to 1.0 with 0.001 granularity. We then compute the convex hull of the resulting 1000 points in the FLOPs-Accuracy plane, This yields the best performing thresholds on the training set. Finally, we use these thresholds on test set and form the FLOPs-Accuracy curves. Each point on a given curve represents one threshold vector.

Table 1: Comparison of E$^2$CM with existing methods under fixed training time budget of six epochs

| Model & Dataset | Method | Acc@0.15 | Acc@0.20 | Acc@0.25 | Acc@0.30 |
|---|---|---|---|---|---|
| ResNet-152 CIFAR-10 | **E$^2$CM + SDN** | **77%** | **85%** | **87%** | **87%** |
| | SDN | 66% | 75% | 85% | 87% |
| | BWDS | 77% | 82% | 84% | 86% |
| | BranchyNet | 57% | 57% | 57% | 57% |
| ResNet-152 CIFAR-10 | **E$^2$CM + SDN** | **84%** | **93%** | **96%** | **96%** |
| | SDN | 63% | 75% | 86% | 94% |
| | BWDS | 84% | 93% | 95% | 95% |
| | BranchyNet | 76% | 77% | 78% | 78% |
| ResNet-152 CIFAR-100 | **E$^2$CM + SDN** | **34%** | **46%** | **52%** | **54%** |
| | SDN | 20% | 24% | 27% | 30% |
| | BWDS | 33% | 40% | 42% | 44% |
| | BranchyNet | 24% | 25% | 26% | 26% |
| WideResNet-101 CIFAR-10 | **E$^2$CM + SDN** | 81% | **87%** | **88%** | **88%** |
| | SDN | 70% | 79% | 86% | 88% |
| | BWDS | **82%** | 85% | 87% | 88% |
| | BranchyNet | 56% | 56% | 57% | 57% |
| WideResNet-101 KMNIST | **E$^2$CM + SDN** | 85% | **92%** | **96%** | **96%** |
| | SDN | 63% | 74% | 96% | 96% |
| | BWDS | **88%** | 90% | 92% | 94% |
| | BranchyNet | 80% | 82% | 83% | 84% |

Points are connected via lines: The performance of any point on any given line is achievable simply via time sharing of the models that correspond to line endpoints.

As shown in Fig. 2, E$^2$CM outperform all existing schemes under a fixed training time budget. Specifically, using class means as the only decision mechanism achieves 35% better accuracy or %50 faster inference time for certain cases. This is because E$^2$CM use the trained weights of the original vanilla model, which generalizes well to the dataset and can be used for classification. On the other hand, internal classifiers require further training. Therefore, they require more time to be ready for the task of classification. Also, it can be seen from Table 1 that combining E$^2$CM with Shallow-Deep Networks (SDN) increases the performance by 50% to 100% when the number of classes is large. Table 1 shows accuracies at 0.15, 0.20, 0.25 and 0.30 FLOPs only because after 0.30 FLOPs, the accuracy stays the same, which indicates that we do not need the tunnel-like design of traditional networks.

It should also be noted that E$^2$CM achieve the same performance as the full network but using only 25%-30% of the available computational resources. This is why Table 1 shows accuracy at up to 0.30 FLOPs. This result strongly suggests that training internal classifiers is not an absolute necessity. Also, E$^2$CM results in a maximum of 9% computational overhead per layer in terms of FLOPs for WideResNet-101 and ResNet-152 on CIFAR-10. Therefore, the amount of extra compute is not a lot. Furthermore, for ResNet-152, WideResNet-101 and ResNet-18 on CIFAR-10, it takes 0.417 MB, 0.43 MB and 0.242 MB of space to store the class means for a layer on average respectively. For ResNet-152 on CIFAR-100, it takes 4.17 MB. Considering that ResNet-152 takes 222 MB and WideResNet-101 takes 476 MB of space, the extra memory usage of E$^2$CM is not a lot.

## 4.2 CLASS MEANS WITH UNLIMITED TRAINING

In this setting, we lift the training time budget. We combine E$^2$CM with Shallow-Deep Networks and compare this combination against BranchyNet, the BWDS method, and Shallow-Deep Networks only.

We train the internal classifiers of our merger of E$^2$CM and Shallow-Deep for 100 epochs. The corresponding high computational complexity may not be desirable for low-power devices. We

Table 2: Comparison of early exit methods

| Model & Dataset | Method | Acc@0.15 | Acc@0.20 | Acc@0.25 | Acc@0.30 |
|---|---|---|---|---|---|
| ResNet-152 CIFAR-10 | **E$^2$CM + SDN** | **82.4%** | **86.4%** | **88.5%** | **88.8%** |
| | SDN | 80% | 86% | 88.4% | 88.8% |
| | BWDS | 82% | 84.1% | 86% | 87.7% |
| | BranchyNet | 80% | 80% | 80% | 80% |
| ResNet-152 KMNIST | **E$^2$CM + SDN** | 94.1% | **96.2%** | **96.8%** | **96.8%** |
| | SDN | 83.9% | 94% | 96.4% | 96.6% |
| | BWDS | **94.5%** | 96% | 96.5% | 96.6% |
| | BranchyNet | 86% | 87.5% | 88.2% | 89.5% |
| WideResNet-101 CIFAR-10 | **E$^2$CM + SDN** | 84% | **88%** | **88.6%** | **88.8%** |
| | SDN | 82% | 88% | 88.5% | 88.8% |
| | BWDS | **85.8%** | 86.5% | 87.8% | 88.2% |
| | BranchyNet | 81.9% | 82.5% | 84.9% | 85% |
| WideResNet-101 KMNIST | **E$^2$CM + SDN** | **94.1%** | **96.5%** | **97.1%** | **97.2%** |
| | SDN | 84.1% | 94% | 96% | 97% |
| | BWDS | 93.8% | 94.2% | 95% | 96% |
| | BranchyNet | 89% | 90.1% | 90.2% | 90.2% |

follow the same threshold selection procedure described above. During inference, the decision of early exit is made according to both the E$^2$CM and the ICs.

We train BranchyResNet-152 and BranchyWideResNet-101 for 300 and 250 epochs respectively. We use stochastic gradient descent with batch size of 256. The loss of the branches are added up to the loss of the final layer and the weighted average is taken as prescribed in Teerapittayanon et al. (2016). We have trained multiple combination of weights, and found that $[1/6, 1/4, 1]$ achieves the best performance. For thresholds, we follow the same procedure, but this time the range of values is $[0, \log K]$ where $K = 10$, because we consider the entropy of the predictions to make a decision.

Since the task of training decision functions in the BWDS method is a binary classification task (i.e., early exit or not) they converge rather quickly. We separately train the classifiers that come after the decision functions as well. We use pooling and single linear layer for these, as suggested by the authors.

As shown in Table 2, combining E$^2$CM with internal classifiers achieves a better trade-off between the computational cost and network accuracy. For low computational budget, E$^2$CM improve the accuracy by more than 2%, with less computation. We also observe that BranchyNet suffers from training the network with the branches jointly. These branches hurt the overall performance. Moreover, convolutional layers in the branches add a significant computational cost without a considerable gain in accuracy.

Although early exit mechanisms with decision functions like the BWDS method provide decent performance, experimental results show that threshold based early exit mechanisms perform better. In other words, making a decision about early exit and then performing classification fares worse than the threshold-based strategies where classification and exiting decisions are melted into the same pot.

According to Table 2, we can conclude that the consulting mechanism between the E$^2$CM and the internal classifiers is generally beneficial. The common decision that is reached by the two classifiers can rectify possible misclassifications and avoid unnecessary computation. This can be seen as an example of ensembles, in which multiple classifiers are used to make a decision. Interestingly, our ensemble reduces the total computational cost unlike ordinary ensemble methods.

The model sizes for ResNet-152 and WideResNet-101 are 222 MB and 476 MB respectively. For the CIFAR-10 dataset, the extra storage overhead of E$^2$CM for ResNet-152 and WideResNet-101 are 22.12 MB and 15.4 MB, which correspond to 10% and 3% of the model sizes, respectively. The overheads are due to the necessity of storing the class means at every layer. These results show that

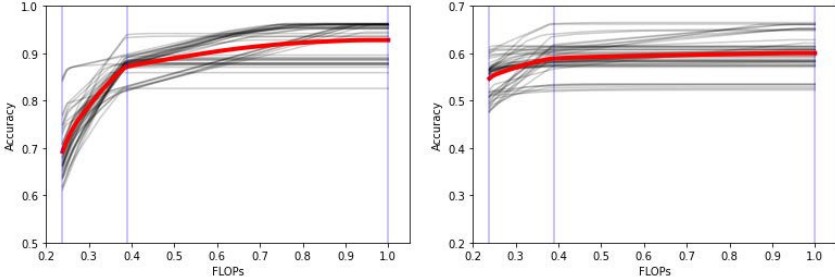

Figure 3: Accuracy-FLOPs curve on MNIST (left) and Fashion-MNIST (right) for unsupervised learning using $E^2CM$. Vertical lines indicate the individual FLOPs of DECs. Each black curve is for one experiment, and the red curve is the average of all experiments.

$E^2CM$ does not have a large memory footprint. Still, one improvement to $E^2CM$ in this context may be to consider the early exits only at specific points as in other existing schemes (Kaya et al., 2019), rather than after each layer. This can reduce the memory footprint as well as computational overhead. However, introducing such modifications give rise to additional hyperparameters, which may be undesirable.

### 4.3 CLASS MEANS FOR UNSUPERVISED LEARNING

We follow the same experimental setup as in Xie et al. (2016) for MNIST (LeCun et al., 1998) and Fashion-MNIST (Xiao et al., 2017) datasets. Namely, we use an encoder with 4 layers, which have 500, 500, 2000, 10 neurons, with a clustering layer (CL) at the end. Let this encoder and clustering layer be $DEC_{large}$. After pretraining $DEC_{large}$, we train the clustering layer as in Xie et al. (2016). Then, we create another encoder with the 500-500-10-CL architecture, where the weights of first two layers are copied from $DEC_{large}$ and frozen. We name this encoder $DEC_{middle}$ and follow the same procedure as in $DEC_{large}$. Finally, we repeat the same procedure for $DEC_{small}$, which has the 500-10-CL architecture, where the weight of the first layer is frozen and copied from $DEC_{large}$.

After the training is complete, we take the intermediate outputs, i.e., the outputs of the layers with 10 neurons. Then, using the cluster centers from each clustering layer, we follow Algorithm 1. To measure the accuracy, we use the same technique described in Xie et al. (2016).

During the experiments, we noticed that the process of pretraining affected the final result significantly. We have thus run multiple experiments, and the performance corresponding to each experiment is illustrated as one gray curve in Fig. 3. The average of individual experiments is shown as the solid red curve.

As seen in Fig. 3 by adding early exits to the architecture, it is possible to save 60% of the computation while losing only 6% in unsupervised clustering accuracy on MNIST dataset. On Fashion-MNNIST, the accuracy loss is 1%. Also, thresholding makes it possible to adjust according to various computational needs.

## 5 CONCLUSION

We propose a novel early exit mechanism based on the class means. Unlike existing early exit mechanisms, our method does not require gradient-based training, which makes it useful for network training on low-power devices. Under fixed training time budget, our method outperforms existing early exit schemes. In addition, combining our method with existing early exit techniques achieve better trade-off between the computational cost and the network accuracy. Moreover, we show that our method is not only useful in supervised learning tasks, but also in unsupervised learning tasks.

ETHICS STATEMENT

Since our technique reduces the inference time with minimal compromise in performance, it can lead to more severe impacts when used in applications in which time has a crucial importance. Stock market prediction and adversarial attacks that are used in the military industry depend heavily on split second decisions, therefore using $E^2CM$ in such applications may cause negative impacts. Also, as in Kaya et al. (2019), new adversarial attacks can be developed to manipulate the computational load of deep learning systems which may hurt performance more than before.

REPRODUCIBILITY STATEMENT

We have uploaded the source code for our method as a supplementary material. For better reproducibility, we have utilized common datasets in our experiments, such as CIFAR-10 (Krizhevsky et al., 2009), KMNIST (Clanuwat et al., 2018), Fashion-MNIST (Xiao et al., 2017) and MNIST (LeCun et al., 1998). We have also used common model architectures such as ResNet-152 (He et al., 2016) and WideResNet-101 (Zagoruyko & Komodakis, 2016).

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
