# OpenReview forum: "E$^2$CM: Early Exit via Class Means for Efficient Supervised and Unsupervised Learning"
_ICLR.cc/2022/Conference — ICLR 2022 Submitted_

### Official Review · Reviewer_Kvv8 · 2021-11-02

**Correctness:** 3
**Technical Novelty And Significance:** 3
**Empirical Novelty And Significance:** 2
**Recommendation:** 3
**Confidence:** 5

**Main Review:**

Strengths:
- The proposed approach is novel, and, to the best of my knowledge, early exits have not been explored from the perspective of non-gradient training, which is a new research direction.
- The empirical evaluation shows that the proposed method does outperform the baselines in a setting with a very limited computational budget. Additionally, the experiments showing how E2CM performs in combination with existing methods are an interesting way to engage with the related work.
- The authors show how their method performs in an unsupervised learning setting, which is an interesting new application of early exits, which were so far mostly considered for classification problems.

Weaknesses:
- The empirical evaluation, in general, is lacking and should be extended, as in the current form the performance and use-cases of the proposed method are not clear. By this, I mean not only considering more complex datasets or different architectures, but also more in-depth analysis of the memory usage and providing statistics to gauge the significance of the results. In particular:
  - The empirical evaluation in the supervised learning setting should be extended. The most complex dataset used in this paper is CIFAR-10, which is currently not considered particularly challenging by the community. The additional results on MNIST variants are not enough to weigh the importance of the method, especially when used with powerful architectures such as ResNet-152 or WideResNet-101. Large-scale experiments on datasets such as ImageNet would be most convincing, but even increasing the scope to SVHN, CIFAR-100 or mini-ImageNet would be beneficial. Since the method does not require editing the base network, would it be possible to adapt it to a wide array of pre-trained models available online (e.g. via websites such as PyTorch Hub or Model Zoo)?
  - The problem of additional memory expenses is not discussed well enough. Although the authors consider the additional memory footprint, they do so for the case of CIFAR-10, where the number of classes is fairly low (10). In practice, many real-world datasets consist of a larger number of classes. For example, for CIFAR-100 the overhead would be 10x higher than for CIFAR-10, which, if I understand correctly, for ResNet-152 means ~100% increase in the memory usage. Similarly, for a dataset with 1000 classes, we have 1000% memory usage overhead. This is quite significant, especially in application to devices with low computational resources. The authors consider adding early exits only at specific points to reduce the memory overhead, but do not show experiments on well it would work in practice.
  - The supervised learning experiments do not include standard deviation - have they been trained with more than one seed?
- The practical motivation for the proposed method is not clear to me. Since the proposed method outperforms other approaches significantly only in the setting with a very limited training budget, the possible applications of the method seem to be restricted. The authors propose some potential applications, but the methods are not tested in settings related to these applications. In particular, the paper claims that "A practical use case for E2CM is when a large, expensive-to-train model is broadcast to edge devices with limited and heterogeneous computation capabilities." (5th paragraph of the Introduction), which I wanted to discuss:
  - "In such a scenario, different devices may train the E2CM model at different FLOP operating points depending on their computation capabilities" - Most early exit methods (e.g. Shallow-Deep Networks considered in this paper) can adapt the FLOP threshold after training by choosing the confidence threshold, denoted by $T_j$ in this paper. So one could train the early-exit model along with the expensive-to-train model, broadcast it to the edge devices, and set the FLOPs as needed.
  - "Another practical scenario for E2CM may be transfer learning, where the training has to be done on a low power edge device with a local dataset that is different from the dataset the base network was trained on" - this use-case is intriguing and more convincing to me, but not tested empirically in the paper and it is not immediately obvious if the method would work (maybe the class means won't be as well separable if we change the domain/task).
  - In the light of application to edge devices, a more thorough discussion of memory usage is even more important.
- The authors do not provide enough theoretical considerations of the method to justify accepting the paper without convincing empirical results. This is not a weakness but itself, but rather highlights the fact that the work has to be judged on the merit of the empirical evaluation, which is lacking.

To sum up, the actionable feedback I propose is providing a more thorough empirical evaluation of the method, clearly stating its limitations, and exploring in-depth the settings the method can be applied to (like transfer learning).

**Summary Of The Paper:**

The paper proposes a method for early-exit-based conditional computation for reducing the inference time of a given neural network. The approach works as follows: E2CM saves the mean activation of each class after each layer and then, during inference, compares the activation of a given example to the recorded mean activations to get a prediction and its certainty. If the certainty is high enough, this prediction is returned, otherwise, the next layer is queried instead.  The authors show empirically that the method outperforms other approaches in a setting with a very limited training budget, and is sometimes able to improve the performance of other methods when applied in combination. Finally, the method is also presented in an unsupervised learning setting where it allows for significant computation savings.

**Summary Of The Review:**

Although the main idea of the paper is intriguing, I think the empirical evaluation needs to be extended and the possible application scenarios should be more thoroughly discussed. In the current version, I cannot recommend accepting this paper for ICLR.

---

> ### Author Response · Authors · 2021-11-22
> **Response to Reviewer Kvv8**
>
> The reviewer suggests extending the supervised learning experiments. We have added the following experiments: ResNet-18 on CIFAR-10 and ResNet-152 on CIFAR-100. As the figures and the tables suggest, our method still outperforms the existing methods. ResNet-18 is a representative model for mobile use-cases, and CIFAR-100 has large number of classes.
>
> The reviewer asks for more discussion on memory usage of E$^2$CM. For ResNet-152 on CIFAR-10, it takes $0.417$ MB of space to store the class means for a layer on average. For WideResNet-101, it t takes 0.43 MB for a layer on average. For ResNet-152 on CIFAR-100, it takes $4.17$ MBs. For ResNet-18 on CIFAR-10, it takes $0.242$ MB. Also, ResNet-152 takes $222$ MB and WideResNet-101 takes $476$ MB of space, which shows that the extra memory usage is very little. This discussion is included in the paper now.
>
> The fourth and fifth paragraphs in the Introduction section explain why it is important to reduce the training and why it might be undesirable to do it on the same hardware platform where the DNN is originally trained. Specifically, circumstances may requires edge devices to dynamically adjust the capabilities of their neural networks over time. For example, when the battery level of an edge device is low, the device can retrain its neural network to a lower computational complexity using E$^2$CM for prolonged operation. Such a training or adaptation requires only one forward pass thanks to E$^2$CM.

---

> > ### Comment · Reviewer_Kvv8 · 2021-11-28
> > **Response to the Authors**
> >
> >
> > >The reviewer suggests extending the supervised learning experiments. We have added the following experiments: ResNet-18 on CIFAR-10 and ResNet-152 on CIFAR-100. As the figures and the tables suggest, our method still outperforms the existing methods. ResNet-18 is a representative model for mobile use-cases, and CIFAR-100 has large number of classes.
> >
> > This is a step in the right direction, but I think a more thorough empirical evaluation is still needed. CIFAR-100 is a good addition, but more settings, in particular those that reflect the applications of your model more directly, would make the empirical evaluation more convincing.
> >
> > > For WideResNet-101, it takes 0.43 MB for a layer on average. For ResNet-152 on CIFAR-100, it takes 4.17MBs. (...) Also, ResNet-152 takes 222MB and WideResNet-101 takes 476MB of space, which shows that the extra memory usage is very little. This discussion is included in the paper now.
> >
> > Correct me if I'm wrong, but since the class activations are 4.17MB per layer on average for ResNet-152 with CIFAR-100, that means in total they take around 633MB or so.  This would means that the base size of the model (222MB) is nearly tripled? This seems like a significant memory overhead, especially in applications with low computational/memory resources.
> >
> > > The fourth and fifth paragraphs in the Introduction section explain why it is important to reduce the training and why it might be undesirable to do it on the same hardware platform where the DNN is originally trained.
> >
> > As mentioned in my original review, authors mention transfer learning as a potential application in paragraphs 4 and 5, but do not provide experiments showing that their method works in this setting.
> >
> > > Specifically, circumstances may requires edge devices to dynamically adjust the capabilities of their neural networks over time.For example, when the battery level of an edge device is low, the device can retrain its neural network to a lower computational complexity using E2CM for prolonged operation. Such a training or adaptation requires only one forward pass thanks to E2CM.
> >
> > If I understand your proposed setup correctly, this can be still be done efficiently by training the model using an early-exit method (e.g. SDN) before pushing it to an edge device, and then changing the confidence threshold (i.e. make the network answer very quickly when the battery is low). As such, I don't think applying E2CM is required here. I raised this point in the original review.
> >
> > Although the experiments section was somewhat improved, the introduced changes are not enough to resolve the overall problems with empirical evaluation. The issues concerning high memory overhead of the method and limited practical applicability remain. As such I decided to keep my score unchanged.

---

> > > ### Author Response · Authors · 2021-11-29
> > > **Response to the comment**
> > >
> > > We would like to thank the reviewer for the response.
> > >
> > > Yes, 4.17MB was per layer for CIFAR100 - We trained our models for 52 early exits - however after thresholding our models use around 8-10 early exits at most. This means the total amount of data to store class means is less than 50MB, which is small compared to the 222MB model. We will provide precise results in this context in the revision.
> > >
> > > Introduction describes different applications where $E^2CM$ may be relevant. We agree that SDN utilizes a similar and practical threshold selection mechanism. A key advantage of $E^2CM$ relative to SDN is that (under the assumption that one trains the same base model from scratch for early exit, such as ResNet152), the training time per epoch remains unchanged in $E^2CM$ thanks to its simple structure. On the other hand, SDN increases training time per epoch by 10%, which may be significant in resource-constrained edge devices. A more general scenario is when an edge device may have its own training data based on which it may want to train a classifier. As the existing simulation results for small number of epochs show, $E^2CM$ is clearly superior in this setup as it can achieve higher accuracy in much less time compared to other methods.
> > >
> > > In general, reducing the training time of neural networks is a fundamental problem with potentially many applications (some of which are mentioned above and in the paper).
> > >
> > > We mentioned transfer learning as one application where $E^2CM$ might be useful, we leave an exploration of this direction as future work.

---

### Official Review · Reviewer_ZDik · 2021-11-02

**Correctness:** 4
**Technical Novelty And Significance:** 2
**Empirical Novelty And Significance:** 2
**Recommendation:** 5
**Confidence:** 4

**Main Review:**

Strengths:
- The paper is well written and easy to follow.
- The proposed method is very simple and intuitive. It can be adopted in bare DNNs or in combination with previous early exit methods.

Weaknesses:
- Since the paper assumes that the DNN is pretrained, the early exit setup, e.g., the training of the sallow nets in (Kaya et al 2019) or the computation of per-class means and setting the threshold, can be done on the same HW platform where the DNN is originally trained. This step can thus be done prior to deploying the DNN in inference mode on a constrained embedded device. As such, it is unclear why it is important to reduce the training overhead for the early exit strategy? The evaluations in the non-constrained training time mode show that the proposed method cannot compete with prior methods unless it is combined with them.
- Regarding the experiments in section 4.1, is the time overhead of finding the optimal threshold for a given FLOP constraint considered within the timing budget? For a fair comparison, the total allocated time to all methods should be equal, rather than just matching the number of forward passes. That being said, it would be great if the authors can comment on my previous question as to why the training budget needs to be limited.
- The evaluated datasets are fairly simple, therefore it is not clear how well the method can generalize in face of more complicated data patterns such as ImageNet. Specifically, since the method depends on per-class means, it is important to evaluate on datasets with more than 10 layers, e.g., CIFAR100, or ImageNet. It is possible that as the number of classes increases, the intermediate representations, particularly in the earlier layers can no longer discriminate between the classes as easily, thereby increasing the FLOPs of this method.


**Summary Of The Paper:**

The paper proposes a method for early exit during inference of DNNs. The method does not require fine-tuning or gradient computation. Rather, it uses the learned embeddings from intermediate model layers to decide where to terminate the execution. Specifically, for a trained model, the method calculates the per-class means of intermediate activation. The distance between the activations of new samples is then computed with the per-class means and inference is exited once the distance becomes lower than a set threshold.

**Summary Of The Review:**

Please see the above review.

---

> ### Author Response · Authors · 2021-11-22
> **Response to Reviewer ZDik**
>
> The fourth and fifth paragraphs in the Introduction section explain why it is important to reduce the training and why it might be undesirable to do it on the same hardware platform where the DNN is originally trained. Specifically, circumstances may requires edge devices to dynamically adjust the capabilities of their neural networks over time. For example, when the battery level of an edge device is low, the device can retrain its neural network to a lower computational complexity using E$^2$CM for prolonged operation. Such a training or adaptation requires only one forward pass thanks to E$^2$CM.
>
> The reviewer asks about the time cost of finding the thresholds. Finding the optimal threshold for a given FLOP constraint is not considered within the timing budget, because this is part of the training procedure.
>
> The reviewer suggests adding more experiments. We have added the following experiments: ResNet-18 on CIFAR-10 and ResNet-152 on CIFAR-100. As the figures and the tables suggest, our method still outperforms the existing methods. ResNet-18 is a representative model for mobile use-cases, and CIFAR-100 has large number of classes.

---

### Official Review · Reviewer_jksT · 2021-11-04

**Correctness:** 3
**Technical Novelty And Significance:** 2
**Empirical Novelty And Significance:** 2
**Recommendation:** 5
**Confidence:** 3

**Main Review:**

I think studying early-exit is a very important research direction and the focus on simplicity is important (if not more than final performance) for achieving real world impact. As far as I am aware using nearest centroid classification (NCC) for early exit classification is novel (though not super original as NCC is used widely for classification) and the experimental study provided can be useful for future research.

This work is solely experimental, and is potentially interesting to the ICLR community. However, the experimental study is limited to some small datasets and lacks in-depth explorations. It is not clear how would the method scale to ImageNet or similar larger datasets/resolutions.

# Concerns
- Even though the authors of [1] focused on NLP, I think their method should be included as a baseline and possibly use the combination method to increase E2CM performance even further. Shallow-deep+e2cm is kind of similar to [1] as it uses 2 decision to decide to skip to the next layer (compared to using 2 decision to stop). Not sure how easy to do this during rebuttal, but regardless this would be a useful addition to the next version.
- Is there a reason why authors choose to experiment with ResNet-152 and WideResNet-101 on Cifar/Mnist instead of repeating experiments done in previous work; let's say Shallow-Deep experiments. Using same benchmarks help researcher compare methods easily and therefore accelerates research. Moreover, Shallow-Deep experiments seems to be more diverse in terms of architecture and data. I recommend authors to have (at-least-some) matching experiments. Imagenet experiments would be appropriate, too; given the relatively small cost of the method.
- I think many of the experimental details can be shared in appendix creating space for some new experiments. It would be nice to see some ablations and in-depth studies in the paper. some ideas: (a) effect of different normalizations: how does different normalizations used in architecture affect results. (b) How about the depth of the network? How does E2CM work on resnet-22? (C) How does the width of the network effects the utility/comparison.
- Authors should give more details on binary search used to choose thresholds. An algorithm box in appendix would be nice. Similarly sharing the final algorithm for Shallow-Deep+E2CM would be nice. Reading the text it is not clear how does shallow-deep+e2cm works. Specially there are 2 ideas (1) concat of class means with activations (2) using 2 stage early-exit strategy. How do you do both of these?
- Authors should compare their class-mean approach with prototypical adaptation (it is different), but readers would appreciate a short discussion on similarities and differences.
- In Figure-2, why is the Branchy net curve flat? Furthermore Branchy-net uses 2 intermedieate activations. Do e2cm and shallow-deep methods use same number of intermediate layers? If e2cm uses all and shallow-deep uses 6, this might not be a fair comparison. Ideally all methods should be able to exit at same layers.

# Minor
- $c_4^kj$ is not defined. Is it a typo?
- Authors mention 1000 points in text, however Figure-2 has much less points for each curve. How do you choose the points? Are they chosen on test set?
- Using different symbols for different curves would make it easier to read in print.
- "One of the primary reasons behind traditional deep learning models’ high computation demand is their tunnel-like design" It is the fact that they bring better performance. Their tunnel like design prevents parallelism, though helps finding more complex functions.
- "One practical scenario for E2CM may be transfer l" Here e2cm is not defined yet. Possibly remove this paragraph.
- "which suggest our method is more agile and powerful yet simple" -> agile and simple.
- "Given a fixed training time budget" limited training time

[1] Bert Loses Patience: https://arxiv.org/abs/2006.04152
[2] https://arxiv.org/pdf/2106.05409.pdf

# After Rebuttal
I read authors' response. I thank authors for sharing their preliminary results on Resnet-56, I think adding rest of the experiments would be a great addition to the paper. Similarly, adding [1] to the experiments would be great (I don't think the algorithm requires any domain specific adjustments, the idea is relatively simple). In short, most of my concerns stay, thus I keep my score. Hope the authors find the feedback useful.

**Summary Of The Paper:**

This paper proposes a new method for performing early exits on a pre-trained architecture. Authors use support set to calculate class prototypes at intermediate representations. Then at evaluation execution on the network is stopped whenever the representations are significantly closer to a class at any level. Results on small datasets (cifar-mnist) using very deep networks show that proposed method achieve better results compared to some of the previous work.

**Summary Of The Review:**

I think studying early-exit inference is a very promising research direction and I appreciate this work due its simplicity, which is important for achieving real world impact. As far as I am aware using nearest centroid classification (NCC) for early exit classification is novel and the experimental study provided can be useful for future research. However the potential impact of this experimental work is currently limited due to its choice of datasets/architectures and lacks in-depth investigations/ablations. Without these, I am concerned that the impact of the work would be limited. For example, it is not clear whether the method would scale to larger datasets and perform better than other alternatives.

---

> ### Author Response · Authors · 2021-11-22
> **Response to Reviewer jksT**
>
> The reviwer suggests comparing E$^2$CM with "BERT Loses Patience: Fast and Robust Inference with Early Exit". All of the experiments stated in this paper is performed on language datasets and the method appears to be tuned for NLP specifically. We believe that extending it to image classification for the purpose of comparing it with E$^2$CM is beyond the scope of our paper.
>
> The reason we have used ResNet architectures in our experiments is that they work well on image classification task. The reviewer suggests including a matching experiment with the Shallow-Deep Networks experiments. We have included the results ResNet-56 experiments below for the training time budget of one epoch.
>
> | Model & Dataset | Method | Acc @0.10 | Acc @0.20 | Acc @0.30 | Acc @0.40 | Acc @0.50 |
> | ----------- | ----------- | ----------- | ----------- | ----------- | ----------- | ----------- |
> | ResNet-56 on CIFAR-10 | E$^2$CM | **0.38** | **0.44** | **0.52** | **0.58** | **0.65** |
> | ResNet-56 on CIFAR-10| SDN | 0.255 | 0.33 | 0.43 | 0.52 | 0.62 | **0.80** |
> | ResNet-56 on CIFAR-100 | E$^2$CM | **0.15** | **0.22** | **0.26** | **0.34** | **0.40** |
> | ResNet-56 on CIFAR-100| SDN | 0.04 | 0.10 | 0.16 | 0.25 | 0.34 |
>
> The reviewer recommends including all of the experimental details in the Appendix. All of the experiment details are given in the Results section. The effect of different architectures can be seen in the figures and tables.
>
> The reviewer mentions that giving an algorithm box for binary search for threshold selection would be beneficial. This is explained in the fourth paragraph of Section 4.1. The final algorithm for the combination of E$^2$CM and Shallow-Deep Networks is described in the third paragraph of Section 4.1.
>
> The reviewer asks about the reason why BranchyNet curve is flat. BranchyNet curve was flat (we now show a table instead of a figure) because the network is trained from scratch under the fixed training time budget, and each exit gives roughly the same performance. To clarify, it now reads ``We also observe that BranchyNet suffers from training the network with the branches jointly. These branches hurt the overall performance. Moreover, convolutional layers in the branches add a significant computational cost without a considerable gain in accuracy.'' in Section 4.2 of the paper.
>
> The reviewer asks about the number of points for the curves. We compute the convex hull of the $1000$ points in the FLOPs-Accuracy plane, which results in fewer points. To clarify, it now reads ``We then compute the convex hull of the resulting $1000$ points in the FLOPs-Accuracy plane, This yields the best performing thresholds on the training set.'' in Section 4.1 of the paper.

---

### Official Review · Reviewer_7dU6 · 2021-11-04

**Correctness:** 3
**Technical Novelty And Significance:** 3
**Empirical Novelty And Significance:** 2
**Recommendation:** 3
**Confidence:** 5

**Main Review:**

I think the idea of the paper is cute, and it surprised me that this worked. A pleasant surprise, and I at least learned something new. For the given setup/data, it also seems to work well. Being able to add early-exiting to any network would be a very welcome addition to any framework that runs neural networks on devices.

However, when reading the paper there are some glaring holes and potential problems with the method that the authors don't comment on.

First of all is the actual computational overhead. This is definitely non-trivial. Take imagenet for example, one would have to do a lot of extra compute for each layer. As for every layer, we'd have to calculate a 1000 feature-map-size subtraction, matrix square and a large sum. Afterwards a softmax is taken, which for large numbers of classes (in e.g. language models this is very relevant), can be expensive as well. The authors seem to not really comment on this, nor calculate the actual overhead per layer. Only in the comparison to other methods do the flops come up. Second is the parameter overhead. The method is not parameter-free. Likely far from it. For every feature map, num_classesfeature_map size parameters have to be stored for comparison. For e.g. Imagenet with a 1000 classes, this is akin to storing a full forward pass of activations of a batch-size=1000. This can run into the gigabytes of memory! Given the author's focus on mobile devices, this doesn't seem very feasible.

The authors seem to avoid this problem by not considering datasets/tasks with a lot of class labels. For MNIST/CIFAR10, your overhead is not going to be too big, but scale this up to language models or sizeable classification models and the overhead is likely prohibitive. The second reason why these drawbacks seem to not show up in the paper is the usage of really really large models for MNIST and CIFAR10. Not only is the usage of these oversized models at odds with the idea of running them on mobile devices, there are far more efficient networks than those mentioned in the paper nowadays, as we can see from figure 1, after 30 layers the network has already classified the examples properly. One could just use that 30 layer network, stick a single softmax at the end, and use that for classification. I wonder what happens to the gains of this method when comparing on such more efficient models.

In terms of comparisons, I'm missing multi-scale dense nets. This is the de-facto paper in this area, and can't be missed from a comparison perspective. I also think the authors could have noted comparison with other methods that do something similar, like methods that drop features, networks that do anytime routing etc. (Adaptive neural networks for efficient inference, Deciding how to decide: Dynamic routing in artificial neural networks, Anytime recognition with routing convolutional network, Resolution Adaptive Networks for Efficient Inference, Skipnet: Learning dynamic routing in convolutional networks)

On a fun note, I wonder what happens if the euclidian distance metric is switched to a cosine one. This would turn the problem into a flattened weight tensor such that we have x.dot(c) for each class. Essentially this is a single neural network layer, with a softmax afterwards. This could work just as well, and make it easier to 'retrain'. Alternatively, you could consider what happens if you had more cluster centers than num_classes. Perhaps you'd be cheaper off mapping the entire feature map to some smaller sub-space, and then projecting back to the softmax output with e.g. an SVD method :)



**Summary Of The Paper:**

The paper introduces a novel mechanism to add early exiting on existing networks without re-training the original network. The paper mentions the reason for this is that this can be done on lower-powered devices on-the-fly, where retraining is not wanted. The method does so by calculating output means, and comparing new examples with that, by running a distance function through a softmax, and early exiting if the prediction is above a certain threshold.

**Summary Of The Review:**

All-in-all, the paper is interesting, but the practicality of the method is very problematic.

I think I could increase my score if:

- The authors commented far more on the added complexity and parameter count
- Had comparisons to methods that have large num-classes
- Had comparisons on networks that were actually representative of mobile use-cases, or at least something that's more reasonably sized than a 150 layer network for CIFAR10
- Compared their method to multi-scale dense nets, and other mentioned methods

---

> ### Author Response · Authors · 2021-11-22
> **Response to Reviewer 7dU6**
>
> For the proposed E$^2$CM scheme, the computational overhead per layer is very low. For example, if a layer costs $1.0$ FLOPs, then E$^2$CM results in a maximum of $0.09$ FLOPs computational overhead per layer for WideResNet-101 and ResNet-152 on CIFAR-10. Therefore, the amount of extra compute is insignificant relative to the overall complexity of the neural network. We have included this note in the paper.
>
> The reviewer claims that E$^2$CM is not parameter-free due to the necessity of storing class means during the inference phase. We would like to note that class means cannot be counted as parameters as they are fixed functions of the classes and the neural network weights. Given a fixed network, the class means are thus fixed and cannot be tuned. In this context, class means should be understood as variables rather than parameters. It is not clear to us how the reviewer came up with his/her calculation of ``gigabytes of memory''. In fact, for ResNet-152, WideResNet-101 and ResNet-18 on CIFAR-10, it takes $0.417$ MB, $0.43$ MB and $0.242$ MB of space to store the class means for a layer on average respectively. For ResNet-152 on CIFAR-100, it takes 4.17 MB. Also, ResNet-152 takes $222$ MB and WideResNet-101 takes $476$ MB of space, which shows that the extra memory usage by E$^2$CM is very little relative to the model size. We have included this note in the paper.
>
> The reviewer suggests we include more experiments for lighter models and for datasets with larger number of classes. We have added the following experiments: ResNet-18 on CIFAR-10 and ResNet-152 on CIFAR-100. As the tables and figures suggest, our method still outperforms existing methods. ResNet-18 is a representative model for mobile use-cases, and CIFAR-100 has large number of classes.
>
> The reviewer suggested comparing E$^2$CM with some papers. We have actually compared E$^2$CM to “Adaptive Neural Networks for Efficient Inference" (BWDS in Table 1). We have mentioned methods that use multiscale inputs/networks in Section 2. We are working on language models at the moment, and we are thinking about extending our method in future for language models. We appreciate the reviewer’s ideas about distance metrics and more cluster centers, and we are going to consider these as future works.

---

> > ### Comment · Reviewer_7dU6 · 2021-11-29
> > **Rebuttal reply**
> >
> > > For the proposed E
> > CM scheme, the computational overhead per layer is very low. For example, if a layer costs  FLOPs, then E2CM results in a maximum of  FLOPs computational overhead per layer for WideResNet-101 and ResNet-152 on CIFAR-10. Therefore, the amount of extra compute is insignificant relative to the overall complexity of the neural network. We have included this note in the paper.
> >
> > This is the case for a standard convolutional layer, with 10 classes. However, this still scales badly both with the amount of classes and anything but a 3x3 convolution. Let's compare convolutions on a single input pixel, with c in and out channels, using mac cost. For a 3x3 convolution, the cost is c^2* 9. For a 1x1 conv c^2. For a depth-wise 3x3 conv, c*9. For E2MC the cost, counting the subtraction, abs, square and comparisons liberally as 1 mac, is the number of classes. Thus if we have e.g. a 64 channeled depth-wise 3x3 conv, and a 1000 classes like in imagenet, the E2MC cost is actually bigger than the layer itself. For 1x1 convs, quite normal in many networks, with a 64 channel-sized layer we have 4096 macs per pixel, whereas E2MC for imagenet is a 1000 macs. That's a 25% overhead. I don't believe this is insignificant, and depth-wise layers and 1x1 layers make up most of the efficient networks ran on mobile devices today.
> >
> > > The reviewer claims that E2CM is not parameter-free due to the necessity of storing class means during the inference phase. We would like to note that class means cannot be counted as parameters as they are fixed functions of the classes and the neural network weights. Given a fixed network, the class means are thus fixed and cannot be tuned. In this context, class means should be understood as variables rather than parameters. It is not clear to us how the reviewer came up with his/her calculation of ``gigabytes of memory''. In fact, for ResNet-152, WideResNet-101 and ResNet-18 on CIFAR-10, it takes  4.17MB,  0.43MB and  0.242MB of space to store the class means for a layer on average respectively. For ResNet-152 on CIFAR-100, it takes 4.17 MB. Also, ResNet-152 takes 222MB and WideResNet-101 takes  476MB of space, which shows that the extra memory usage by E2CM is very little relative to the model size. We have included this note in the paper.
> >
> > I don't think arguing semantics on variables and parameters here is changing my point. The point is related to the amount of memory read-writes you'd have to do with this method. No matter if the values are trained or not, memory is a significant limiting factor in both efficiency and energy performance on mobile devices. Increasing model sizes by a factor 4, would in many cases lead to slow-downs or energy consumption multiplying by 4 as well. As the other author argued, for resnet-152 on cifar-100 it takes 4.17 MB per layer. That per-layer, means the model size would still be roughly tripled. This is way too much overhead for a practical on-device method. This would get even worse on Imagenet.
> >
> > > The reviewer suggests we include more experiments for lighter models and for datasets with larger number of classes. We have added the following experiments: ResNet-18 on CIFAR-10 and ResNet-152 on CIFAR-100. As the tables and figures suggest, our method still outperforms existing methods. ResNet-18 is a representative model for mobile use-cases, and CIFAR-100 has large number of classes.
> >
> > I definitely appreciate the extra results, but I would personally really look for: ImageNet-sized experiments. Experiments with actual mobile-relevant models like mobilenetv3/efficientnet based models. For the ResNet18 model, I would also like to see where the early-exiting happens and is useful. Sadly, my statement regarding the 'take a resnet-152 network, cut-off the last 120 layers and train a single softmax on that' as a simple baseline experiment went unmentioned :(.
> >
> > All-in-all, after reading the author's feedback, and the other reviewer's feedback, I still think the main faults of this method hold, and make it a non-practically applicable paper. Despite the interesting insight that early-exiting can be done effectively based on per-layer information early on in common networks.
> > There is a significant overhead in terms of compute and memory for efficient mobile-use-cases, which are generally memory starved. Since, as the other reviewer noted, the method generally works better than others only in small-training mobile-oriented regimes, the method is as odds with itself. Both being only applicable in slightly arcane practical scenarios, and not working well on those due to the overhead.
> >
> > For this reason I am retaining my score of a reject. It is important that the readers of papers at this conference spend their time reading something that will either work for them in practice and/or give them a useful insight to base the rest of their own future work on.

---

> > > ### Author Response · Authors · 2021-11-29
> > > **Response to reviewer**
> > >
> > > We would like to thank the reviewer for their comments.
> > >
> > > The revised version of the paper contains experiments for CIFAR-100, which is a 100-class example. The example shows that the low computational complexity or other benefits of E2CM remain. We have not yet performed experiments on a 1000-class dataset, but we expect similar results. Suppose for the sake of argument that the method performs perfectly well for the 1000-class example as well. The issue is that one can then question the practicality of the method for a million-class example, reasoning that the method overhead scales as O(N), where N is the number of classes. Every ML method ever proposed can be pushed to the point of impracticality in a similar manner by an appropriate choice of the training/testing environment...
> > >
> > > We believe that the range of parameters, number of classes, etc are within practical ranges (where they will be useful under certain conditions) and E2CM will thus be useful for such scenarios. For example, we believe that 100 classes is a reasonable large number for many practical applications. Scaling E2CM to a million-class example is a direction for future research (whether one can reduce O(N) to O(log N) for example). Also, competing methods such as BranchyNet and Shallow-Deep consider a similar maximum number of classes (10 and 200 respectively), so we are improving the state of the art. Nevertheless, we plan to incorporate ImageNet experiments in a future revision of the present work.
> > >
> > > About memory overhead: We did not necessarily want to argue about semantics but wanted to emphasize parameter-free as that means less number of parameters to train or tune, which is an advantage for any method. 4.17MB was per layer for CIFAR100 - We trained our models for 52 early exits - however after thresholding our models use around 8-10 early exits at most. This means the total amount of data to store class means is less than 50MB, which is small compared to the 222MB model for a 100-class example. We will provide precise results in this context in the revision.
> > >
> > >  'take a resnet-152 network, cut-off the last 120 layers and train a single softmax on that' This is essentially what methods like shallow-deep networks do for the case of a single exit. The fact that we provide examples for 6 exits for shallow-deep means that we have this case as well as a special case.

---

### Decision · Program_Chairs · 2022-01-20

**Decision:**

Reject

**Comment:**

This paper proposes an early exit method that uses class means of samples that is gradient free and is aimed for low compute cases such as mobile and edge data. The idea is novel in this setting (though class means have been used for other settings such as few shot classification) and empirical results show that it works well. There are two main concerns from reviewer concerns that were not addressed by the author rebuttal. First, applicability of the model in real world due to its memory requirements and two, experiments that show performance on more realistic datasets such as Imagenet. The reason the latter is required is the promise of mobile application for the proposed method. I suggest the authors explain the first concern more and add the requested experiments in the upcoming version of the paper.